# Cognitive Dysfunction in Non-Alcoholic Fatty Liver Disease—Current Knowledge, Mechanisms and Perspectives

**DOI:** 10.3390/jcm10040673

**Published:** 2021-02-09

**Authors:** Kristoffer Kjærgaard, Anne Catrine Daugaard Mikkelsen, Charlotte Wilhelmina Wernberg, Lea Ladegaard Grønkjær, Peter Lykke Eriksen, Malene Flensborg Damholdt, Rajeshwar Prosad Mookerjee, Hendrik Vilstrup, Mette Munk Lauridsen, Karen Louise Thomsen

**Affiliations:** 1Department of Hepatology and Gastroenterology, Aarhus University Hospital, Palle Juul-Jensens Boulevard 99, C117, 8200 Aarhus, Denmark; ancami@clin.au.dk (A.C.D.M.); ple@clin.au.dk (P.L.E.); r.mookerjee@ucl.ac.uk (R.P.M.); vilstrup@clin.au.dk (H.V.); karethom@rm.dk (K.L.T.); 2Department of Gastroenterology and Hepatology, Hospital of South West Jutland, Finsensgade 35, 6700 Esbjerg, Denmark; charlotte.wilhelmina.wernberg@rsyd.dk (C.W.W.); lea.ladegaard.gronkjaer@rsyd.dk (L.L.G.); mette.enok.munk.lauridsen@rsyd.dk (M.M.L.); 3Institute of Psychology and Behavioural Sciences, Aarhus University, Bartholins Allé 11, 8000 Aarhus, Denmark; malenefd@psy.au.dk; 4Department of Clinical Medicine, Aarhus University, Palle Juul-Jensens Boulevard 82, 8200 Aarhus, Denmark; 5Institute for Liver and Digestive Health, University College London, Rowland Hill Street, London NW3 2PF, UK

**Keywords:** non-alcoholic steatohepatitis, cognition, inflammation, ammonia, vascular dysfunction, neurodegeneration, neuropsychology, psychometric, hepatic encephalopathy

## Abstract

Non-alcoholic fatty liver disease (NAFLD) has emerged as the hepatic component of the metabolic syndrome and now seemingly affects one-fourth of the world population. Features associated with NAFLD and the metabolic syndrome have frequently been linked to cognitive dysfunction, i.e. systemic inflammation, vascular dysfunction, and sleep apnoea. However, emerging evidence suggests that NAFLD may be a cause of cognitive dysfunction independent of these factors. NAFLD in addition exhibits dysbiosis of the gut microbiota and impaired urea cycle function, favouring systemic ammonia accumulation and further promotes systemic inflammation. Such disruption of the gut–liver–brain axis is essential in the pathogenesis of hepatic encephalopathy, the neuropsychiatric syndrome associated with progressive liver disease. Considering the growing burden of NAFLD, the morbidity from cognitive impairment is expected to have huge societal and economic impact. The present paper provides a review of the available evidence for cognitive dysfunction in NAFLD and outlines its possible mechanisms. Moreover, the clinical challenges of characterizing and diagnosing cognitive dysfunction in NAFLD are discussed.

## 1. Introduction

Non-alcoholic fatty liver disease (NAFLD) is the hepatic component associated with the metabolic syndrome and constitutes a major global health burden, affecting an alarming one-fourth of the general population worldwide [1,2,3]. NAFLD comprises the progressive disease spectrum from simple steatosis through non-alcoholic steatohepatitis (NASH), with or without fibrosis, to cirrhosis [4,5,6,7]. Morbidity and mortality are related to both the liver disease itself and to extrahepatic complications associated with NAFLD and the metabolic syndrome, in particular cardiovascular disease [8,9,10,11,12,13]. In recent years, also cognitive dysfunction has been increasingly recognized as a complication of NAFLD [14,15] as problems with memory, attention, concentration, forgetfulness, and confusion have been reported in up to 70% of NAFLD cases with associated negative impact on everyday living and quality of life [16,17,18,19].

The metabolic syndrome is defined by the presence of abdominal obesity, peripheral insulin resistance, hypertension and dyslipidaemia [20]. Several features of the metabolic syndrome, i.e., systemic inflammation, vascular dysfunction, atherosclerosis and obstructive sleep apnoea (OSA), have frequently been linked to cognitive disturbances, which has given rise to the concept of the metabolic cognitive syndrome [21]. These are all features also linked with NAFLD [22,23], but it is unclear if NAFLD in itself gives rise to and contributes to cognitive dysfunction. Adding to these features, NAFLD exhibits disruption of the gut microbiota and impairment of urea synthesis in the liver, leading to ammonia accumulation even in precirrhotic stages [24,25]. These disturbances, compounded by systemic inflammation, are central elements of the gut–liver–brain axis and acknowledged as significant in the pathogenesis of hepatic encephalopathy (HE), the neuropsychiatric syndrome associated with progressive liver injury [26,27,28,29].

Considering the high prevalence of NAFLD, its potential adverse impact on cognitive function represents a clinical challenge with expected huge societal and economic consequences. The present paper provides a review of the current evidence for NAFLD cognitive dysfunction and outlines the possible mechanisms involved in the development of brain dysfunction in NAFLD. Moreover, the clinical challenges of characterization and diagnostic testing are discussed.

## 2. Evidence for Cognitive Dysfunction in NAFLD

The existing literature on cognitive dysfunction in NAFLD is outlined in Table 1. Cognitive function in NAFLD has only been investigated on a larger scale in three population-based observational studies. The first comprehensive study was undertaken by Seo et al. using data from the 1988–1994 National Health and Nutrition Examination Survey (NHANES), comprising 874 NAFLD patients and 3598 healthy controls below the age of 59 years [30]. Here, NAFLD was associated with poor memory and attention (serial digit learning task; SDLT), independent of important confounders. In addition, NAFLD patients showed deficits in visuospatial function (digit symbol substitution test; DSST) and psychomotor speed (simple reaction time test; SRTT). However, the latter deficits were not significant after adjustment for life-style related confounders. Weinstein et al. (2018) used NHANES data from 2011 to 2014, comprising 1102 subjects above 60 years of age whereof 413 were diagnosed with NAFLD [31]. In this study, a lone NAFLD diagnosis was not associated with poor performance on any of the cognitive tests, whereas NAFLD with concurrent type 2 diabetes mellitus (T2DM) was associated with impaired visuospatial function (DSST). In fact, this group performed significantly worse than all other groups, including T2DM alone, suggesting that NAFLD adds to the cognitive dysfunction in T2DM that has frequently been reported [32]. Finally, Weinstein et al. (2019) studied cognitive function in 1278 subjects of which 378 had a NAFLD diagnosis, using data from the Framingham Heart Study [33]. Overall, NAFLD was not independently associated with cognitive dysfunction. However, a subgroup of NAFLD patients with high risk of having hepatic fibrosis (measured as the NAFLD fibrosis score (NFS)) exhibited signs of impaired executive function (Trailmaking A–B) and abstract reasoning (Wechsler adult intelligence tests-revised (WAIS-R) similarities test; SIM), compared to those with low risk. Smaller cross-sectional studies have found that patients with NAFLD underperform when tested with general dementia screening tools such as the mini mental state examination (MMSE) and Montreal cognitive assessment (MoCA), however, most of these studies were too small to adjust for confounders (Table 1) [34,35,36,37].

The abbreviations for the neuropsychological tests applied in the separate articles are provided in parenthesis. The individual tests are described with references in Appendix A.

One major limitation of the above studies is the lack of biopsy confirmation. Instead, NAFLD was diagnosed by combinations of ultrasound, computed tomography (CT) imaging and various fibrosis/fatty liver scores [30,31,33]. Accordingly, it is unknown how the severity of NAFLD, e.g., steatohepatitis or hepatic fibrosis, impacts cognitive function; based on the studies’ inclusion criteria, patients with more advanced liver disease most likely comprised only a smaller fraction of the NAFLD groups [1]. Another overall limitation is the use of crude neuropsychological screening tools as the majority of tests used are developed for the diagnosis of dementia and may not be sensitive towards the cognitive phenotype of NAFLD.

In conclusion, the studies discussed above do not provide sufficient evidence that the whole spectrum of NAFLD disease is independently associated with cognitive dysfunction. However, cognitive performance seems correlated with liver disease severity in NAFLD [30,33,34,36], but only a few smaller studies have investigated the impact of NAFLD severity and inflammation on cognitive function. Felipo et al. showed that patients with simple steatosis were not cognitively impaired, whereas NASH patients (non-cirrhotic) with hyperammonemia and systemic inflammation performed poorly on all subtests of the portosystemic encephalopathy (PSE) test [38]. This indicates that simple steatosis as such may not be an independent risk factor for cognitive dysfunction and that factors associated with more severe degrees of disease, such as high ammonia levels and systemic inflammation, may be required for cognition to be affected [38].

## 3. Possible Mechanisms behind Cognitive Dysfunction in NAFLD

NAFLD is indeed a multisystem disease, and several clinical features of NAFLD have been linked to cognitive disturbances. We discuss below important features of NAFLD and how these act as possible mechanisms contributing to cognitive dysfunction (Figure 1).

### 3.1. Systemic- and Neuroinflammation

A common feature of NAFLD is chronic low-grade inflammation [23,42,43,44]. Whilst this may start at local tissues such as the adipose tissue and the liver with the presence of hepatic inflammation, i.e. NASH, the inflammation spreads to become systemic and affect other organs, including the brain.

Obesity and insulin resistance perpetuate adipose tissue inflammation and lead to ectopic lipid accumulation in various organs, including the liver. In NAFLD, hepatic fat infiltration and hepatocyte injury stimulate the secretion of chemokines from hepatocytes and non-parenchymal cells to induce recruitment of macrophages to the liver [42,45]. Upon infiltrating the liver, macrophages produce copious amounts of proinflammatory cytokines, which facilitates NASH progression [45,46]. The proinflammatory cycle of macrophage recruitment to the liver is sustained in a feed forward loop by secretion of cytokines and chemokines to the systemic circulation contributing to the development of a systemic, persistent, low-grade inflammation [47,48,49,50]. Systemic inflammation has been proposed as causal for cognitive dysfunction in NAFLD and the metabolic syndrome, but it is unclear how the inflammation spreads to the brain [51,52]. The central nervous system (CNS) is functionally separated from the systemic circulation by the blood–brain barrier (BBB); the neuroimmune system is composed primarily of resident glial cells, whereas the normal integrity of the BBB withholds peripheral immune cells and modulators [53,54]. However, cytokines and other proinflammatory factors are able to bypass the BBB in a number of ways to disturb brain function. They may enter the brain by active transport or through direct entry in circumventricular regions, which are unique points of communication between blood, brain parenchyma and cerebrospinal fluid (CSF), characterized by the lack of BBB [55,56,57]. Moreover, cytokines can activate their receptors on peripheral endothelial cells, promoting the release of proinflammatory factors inside the CNS [58]. 

Finally, neuroinflammation can be induced by chemokine-mediated recruitment of monocytes and other immune cells that transmigrate through the BBB and infiltrate the brain, as seen in CNS infections and brain inflammatory disorders [59].

These processes ultimately result in the activation of microglia, the innate immune cells of the CNS, thereby promoting the release of proinflammatory cytokines and initiation of a complex immune response in the brain [60]. An increasing number of experimental studies have shown evidence of cognitive dysfunction and neuroinflammation in animal models of NASH. Collectively, these studies indicate that NASH is associated with impaired spatial learning and memory as well as depression-like behaviour, and that these neurobehavioral changes are accompanied by microglia activation and increased release of proinflammatory cytokines in brain tissue [61,62,63]. Furthermore, the induction of NAFLD in models of Alzheimer’s disease (AD) and haemorrhagic brain injury exacerbates cognitive impairment and neuroinflammation [64,65]. In human NAFLD, chronic neuroinflammation was suggested by Balzano et al., who studied post-mortem cerebellums of NASH patients and found neuronal loss, lymphocyte infiltration and increased activation of microglia and astrocytes, as compared with control autopsies from subjects without liver or brain disease [66,67].

### 3.2. Metabolic Liver Dysfunction and Ammonia

Emerging evidence shows that NAFLD is associated with impaired metabolic liver function, even in early stages of disease [24,68,69,70]. Urea synthesis occurs exclusively in the liver, constituting the primary pathway for nitrogen (ammonia) elimination through conversion of excess amino-nitrogen to urea [71]. This process is severely impaired in patients with cirrhosis due to loss of hepatic functional mass, leading to systemic accumulation of ammonia [71,72]. However, deficient ureagenesis also appears to be a specific metabolic defect in precirrhotic NAFLD.

Our group has previously shown that both experimental and clinical NAFLD produce a reduction in the in vivo functional capacity for urea synthesis and the expression and function of urea cycle enzymes already at the stage of simple steatosis, resulting in decreased ammonia elimination and ultimately hyperammonemia at a non-cirrhotic stage [24,73,74,75]. Moreover, dietary intervention restored abnormal urea cycle enzyme activity in experimentally induced NASH, accompanied by a decrease in liver fat [74]. Evidence has long suggested that the presence of steatosis adversely affects mitochondrial liver function [76], including the urea cycle, where in vitro studies demonstrated that the accumulation of lipids in hepatocytes and liver tissue slices reduced the gene expression of urea cycle enzymes, and increased ammonia production [74,77]. Thus, steatosis in early NAFLD is likely to be causal for urea cycle dysfunction rather than a mere association [78].

Ammonia is a neurotoxic molecule that readily crosses the BBB and is central in the pathogenesis of HE [26]. However, the pathogenesis of HE is complex and whilst ammonia holds an irrefutable, key role, it is not solely responsible for the neurocognitive dysfunction [29]. Many studies clearly demonstrate that the cognitive manifestations of hyperammonemia can be exacerbated in an inflammatory environment [79,80,81]; it has been proposed that hyperammonemia is the key that “unlocks” the BBB, making the brain susceptible to systemic inflammatory responses [82]. So, systemic inflammation acts synergistically with the deranged nitrogen metabolism found in patients with progressive liver dysfunction to propagate the clinical picture of HE. It is generally accepted that HE represents a primary gliopathy due to astrocyte swelling and oxidative stress, which disturbs astrocytic-neuronal communication, synaptic plasticity and oscillatory networks in the brain, to finally trigger the clinical HE symptoms [26,83]. Even in the absence of clinically overt HE, low-grade astrocyte swelling, as may be seen in NAFLD [66], could have significant functional consequences and impair the cross-talk between swollen astrocytes and neurons [84]. Neuroinflammation is a recognized feature of HE, and microglia activation has repeatedly been demonstrated in response to experimental hyperammonemia and in patients with HE, which contributes to the generation of oxidative stress [28,85,86,87]. In addition, recent evidence suggests that cognitive impairment and neuroinflammation caused by experimentally induced chronic hyperammonemia could be mediated by induction of systemic inflammation that is reversible by reducing systemic ammonia, or by anti-inflammatory treatment [88].

In NAFLD, little is known about the interplay between ammonia and cognitive dysfunction. Felipo et al. demonstrated that the combination of hyperammonemia and chronic, low-grade systemic inflammation in patients with non-cirrhotic NASH was associated with neuropsychiatric disturbances that are commonly associated with hyperammonemia in the context of cirrhosis [38]. In another study, neurobehavioral changes were present in rats with diet-induced NASH, which were accompanied by systemic hyperammonemia and altered concentrations of neurotransmitters in several brain regions [89]. Findings of impaired ureagenesis and hyperammonemia in precirrhotic NASH raise the important question as to whether the cognitive dysfunction observed in NAFLD patients may be considered as a form of minimal HE? It is possible that the chronic, low-grade systemic inflammation in NAFLD could act synergistically with subtle hyperammonemia to disturb cognitive function in early stages of liver disease.

### 3.3. Disturbed Gut Microbiota

A dysfunctional gut–liver–brain axis is associated with HE in patients with cirrhosis [27,90], and the association is clearly illustrated by the beneficial effects on cognition of microbiota alteration with lactulose, rifaximin or faecal microbiota transplantation [91,92]. This knowledge has led to an interest in the gut–liver–brain axis also in non-cirrhotic liver disease.

In precirrhotic stages of NAFLD, gut microbiota is disturbed with increased prevalence of small intestinal bacterial overgrowth, low bacterial richness and an increased ratio between the two phyla, *Firmicutes* and *Bacteroides* [93,94,95]. These microbial characteristics are drivers of a key element in NAFLD pathophysiology, the “leaky gut”, where tight junctional proteins anchoring intestinal endothelial cells lose part of their sealing effect, thereby increasing mucosal permeability [96,97]. The damaged intestinal barrier allows metabolites and bacterial fragments to reach the liver through the portal vein, thereby initiating a cascade of hepatic inflammation, lipogenesis, oxidative stress, insulin resistance and fibrogenesis [98,99,100]. Further, it has been hypothesized that bacterial products from the gut, including endotoxins, ammonia and bacterial DNA initiate and propagate systemic- and neuroinflammation [27,101].

Evidence suggests that patients with NAFLD exhibit both cognitive dysfunction and altered gut microbiota, but no human studies have convincingly linked the two as for cirrhosis. However, a few preclinical studies have examined the pathophysiological aspects of gut microbiota and impaired cognition in NAFLD. One study found intestinal gut dysbiosis and decreased production of gut microbial short-chain-fatty acids in a rat model of NASH, which was associated with neurobehavioral dysfunction and hyperammonemia, as previously mentioned [89]. In another study, control and diet-induced NASH rats were treated with the probiotic *Lactobacillus plantarum* for 2 weeks, and changes in cognitive function and hippocampal histology were investigated [102]. Here, NASH was associated with a decrease in spatial learning and memory, which was reversed by the probiotic treatment. Moreover, an increase in viable cells of the hippocampal region and resolution of hepatic steatosis, ballooning and fibrosis were found in the NASH group after probiotic treatment. The authors did not analyse taxonomy changes in faeces but concluded that the effects of the probiotic treatment were induced by a beneficial change in the gut microbiota. The above findings indicate a significant association between disturbance of the intestinal microbiota and cognitive dysfunction in NAFLD but do not clarify whether this interaction is mediated through the gut–liver axis or through direct effects on the brain.

### 3.4. Atherosclerosis and Cerebrovascular Dysfunction

NAFLD is closely related to vascular complications and associated with a 64% increased risk of fatal or non-fatal cardiovascular events such as myocardial infarction [13]. This association is predominantly owed to the shared metabolic risk factors of NAFLD and cardiovascular disease including hypertension, obesity, insulin resistance and physical inactivity [103], all conditions also associated with cerebrovascular dysfunction [104]. However, NAFLD also acts as an independent risk factor for cardio- and cerebrovascular disease. Hence, NAFLD is independently associated with subclinical atherosclerosis including carotid intima-media thickness, coronary calcification, endothelial dysfunction and arterial stiffness [105], assumably due to hepatic inflammation, which promotes a rise in systemic proinflammatory and procoagulant factors, and increases the risk of atherosclerosis [103,106,107,108].

In NAFLD patients without overt cardiovascular disease, high levels of liver fat detected by CT have been shown to correlate with measures of microvascular hemodynamic dysfunction [109]. Such changes increase the risk of silent subcortical infarcts, witnessed as white matter lesions (WMLs) on magnetic resonance imaging (MRI), which are associated with increased risk of stroke and dementia [110,111,112,113]. However, in a large study of 766 individuals, Weinstein et al. found no difference in the presence of WMLs between NAFLD and healthy controls [114]. This was reproduced in another study of 79 patients with biopsy-proven NAFLD, but interestingly, the number of patients with WMLs was almost doubled in NASH vs. non-NASH groups [115]. The difference in WMLs was even more pronounced when comparing patient groups of moderate to severe fibrosis (F2–4) with no or mild fibrosis (F0–1), demonstrating that F2–F4 fibrosis was independently associated with the number of WMLs.

Tuttolomondo et al. investigated the association between vascular dysfunction and cognitive dysfunction in NAFLD [34]. Patients with biopsy-proven NAFLD were found to have increased arterial stiffness and endothelial dysfunction and lower MMSE scores when compared with healthy controls. Endothelial dysfunction and MMSE were independently associated with NAFLD after adjustment for additional metabolic risk factors, and NAFLD patients with fibrosis had higher levels of arterial stiffness than those without fibrosis. Such vascular alterations in NAFLD may lead to decreased cerebral blood flow (CBF), also important for cognitive function [116]. In a large study of 505 individuals, NAFLD was associated with lower total and grey matter CBF after adjusting for cardiovascular risk factors, but not after adjustment for BMI [117]. Findings of reduced CBF have been reproduced in smaller studies of biopsy-proven NAFLD [118], and one particular study found NAFLD patients to have impaired upregulation of cerebral blood volume measured by oxygenated haemoglobin concentration during a cognitively challenging verbal fluency task [39].

Thus, current evidence strongly suggests that NAFLD is independently associated with vascular dysfunction, resulting in cerebrovascular complications. Importantly, chronic cerebrovascular dysfunction and impaired CBF may eventually induce permanent neurodegenerative changes of the brain. While the link between microvascular disease in the brain and cognitive decline is firmly established, studies assessing the impact of atherosclerosis and cerebrovascular dysfunction on cognitive impairment in NAFLD patients are warranted.

### 3.5. Neurodegeneration

The metabolic syndrome and its components of T2DM and obesity are increasingly recognized as important in the pathogenesis of neurodegeneration and are established risk factors for the development of both mild cognitive impairment (MCI) and dementia [119,120,121,122,123]. As described, vascular dysfunction in NAFLD is associated with an increased risk of cerebrovascular complications causing neurodegenerative changes of the brain, and as such, the dementia risk burden of NAFLD is driven primarily by vascular forms. Despite the recent attention to cognitive dysfunction in NAFLD, studies investigating the association between NAFLD and dementia are scarce.

One population-based study found that NAFLD was independently associated with a 16% increased risk of overall dementia (HR 1.22 for vascular dementia, HR 1.07 for AD) [124]. Another population-based study found that physically frail NAFLD patients with high risk of liver fibrosis were at a 5-fold increased risk of receiving a dementia diagnosis over a 8-year period, while physical frailty nor high risk of liver fibrosis were independent risk factors on their own [125]. Conversely, a third study found NAFLD not to be associated with a dementia diagnosis over a 10-year period in elderly patients ≥ 65 years [126]. In MRI brain studies, NAFLD was independently associated with reduced total brain volume, corresponding to 4.2 years of brain aging when compared with age-matched controls, as estimated by Weinstein et al. [114,117]. While these studies did not investigate cognitive function, regional atrophy on MRI has previously been shown to precede MCI in cognitively normal individuals and thus suggests a link between NAFLD and neurodegeneration [112]. Concordantly, in a small study, Filipovic et al. showed that reduced brain volume in patients with NAFLD was associated with poor cognitive performance [35]. In a juvenile minipig model of diet-induced NAFLD, neurodegeneration was demonstrated as neuronal loss and astrogliosis in the frontal cortex, compared with control pigs [127].

Interestingly, studies suggest that NAFLD also increases the risk of AD, and pathways related to metabolic dysfunction are in fact emerging as important drivers of AD pathology. A recent network clustering analysis showed that NAFLD and AD share 189 genes, divided in three major pathways: carbohydrate metabolism, long fatty acid metabolism and IL-17 signaling pathways [128]. It is now known that AD is associated with global reductions in cerebral glucose metabolism, and the crucial role of the insulin/IGF-I pathway in neuronal cell function has led to the hypothesis of AD as a “diabetes type 3” [129]. This phenomenon can arise as secondary to peripheral insulin resistance, which is present in NAFLD [130]. Moreover, hepatic steatosis promotes lipolysis and generation of toxic sphingolipids in the liver including ceramides that are mediators of insulin resistance and inflammation able to cross the BBB [129,131]. Finally, the functional low-density lipoprotein-related protein (LRP-1), which is abundantly expressed in hepatocytes and sinusoidal cells, facilitates the clearance of circulating amyloid β proteins and thereby plays an important role in AD progression and cognitive function [132,133]. It has been shown that liver dysfunction and hepatic insulin resistance impedes clearance of circulation amyloid β through a reduction in hepatic LRP-1 expression and LRP-1 translocation to the hepatocyte membrane [134,135]. The role of NAFLD in AD has been further substantiated by recent studies in murine models of AD, where induction of NAFLD accelerated pathological AD signs such as neuroinflammation, cerebral hypoperfusion, reduced expression of brain LRP-1, and dysregulation of amyloid β metabolism [64,136]. In summary, hepatic and metabolic dysfunction as present in even early stages of NAFLD contributes to the development of neurodegeneration and could leave a large number of patients at risk of developing dementia.

### 3.6. Obstructive Sleep Apnoea

Given the natural history of disease sharing important risk factors inherent in the metabolic syndrome, NAFLD is associated with obstructive sleep apnoea (OSA) and affects up to 10% of NAFLD patients [137,138,139]. OSA is a disorder characterized by obstructive apnoeas, hypopneas and/or respiratory effort-related arousals caused by repetitive collapse of the upper airway during sleep [140]. The nocturnal hypoxia and disruption of natural sleeping pattern in OSA can induce or worsen inattention, loss of memory and cognitive deficits, which combined may result in impaired executive function [141,142,143]. As for NAFLD, individuals with sleep-disordered breathing are more likely to develop clinically relevant cognitive decline or dementia [144]. However, these changes are more likely to be mediated by the direct effects of nocturnal hypoxia, oxidative stress and endothelial dysfunction, rather than metabolic disease, as is the case for NAFLD. Interestingly, the intermittent hypoxia in OSA has been proposed to drive liver injury in NAFLD [145]; in murine NAFLD, exposure to chronic intermittent hypoxia exacerbates liver inflammation and fibrosis along with systemic inflammation, while OSA and NAFLD severity are independently correlated in morbidly obese patients undergoing bariatric surgery [146,147,148]. Considering the major impact of OSA on brain health, OSA is a potential facilitator of the early cognitive deficits observed in NAFLD patients, in particular for the morbidly obese.

## 4. Methods Used to Characterize Cognition in NAFLD

The neuropsychiatric complications of NAFLD have only recently become a topic of clinical and scientific interest, and accordingly, the cognitive hallmarks of this metabolic encephalopathy are not yet well-defined in human studies. Therefore, it is yet unclear what tests strategies should be pursued to characterize and detect the specific cognitive deficits caused by NAFLD. The challenge in assessing cognitive function in NAFLD is that components of NAFLD and the metabolic syndrome affect cognition independently, but collectively induce metabolic encephalopathy. Thus, factors such as liver dysfunction, vascular dysfunction, sleep apnoea and diabetic microvascular disease may altogether influence cognition to various degrees in the same individual. Patients with NAFLD most likely constitute a cognitively heterogeneous group, and thus efforts should be invested in characterizing the cognitive profile in well-defined subgroups of patients to attain full understanding of this complex condition.

A few human studies have sought to objectively characterize cognitive deficits in NAFLD using various neuropsychological tests on a scale that allows adjustment of important confounders [30,31,33,34,36]. These neuropsychological tests and the cognitive domains that they assess are outlined in Table 1 and further described in Appendix A. As earlier discussed, lack of biopsy confirmation, and inconsistencies in NAFLD inclusion criteria complicate the interpretation and comparison of these studies. Moreover, the test strategies used were somewhat heterogeneous, but in some cases, similar test modalities were applied which allows for a cautious comparison. Problems with attention and memory were found by Seo et al. using a simple test battery consisting of a simple visual-motor reaction time, symbol digit substitution test and serial digit learning test. In another large study, Weinstein et al. (2018) did not find any cognitive deficits specifically in NAFLD patients [31], whereas Weinstein et al. (2019) showed that NAFLD patients with high risk of fibrosis (NAFLD fibrosis score) had difficulties with abstract reasoning (SIM) as well as executive function using the Trailmaking test [33], which is also included in the PSE battery used to diagnose minimal hepatic encephalopathy (MHE) [38]. The studies by Celikbilek and Tuttolomondo found more diverse deficits, i.e., impaired executive and visuospatial function using the simple dementia screening tools MoCA and MMSE, respectively [34,36]. In MHE, problems with psychomotor speed, attention, and executive and visuospatial functions dominate the picture, while memory is not affected [149]. Memory is, however, commonly affected in diabetic encephalopathy, dementia and sleep apnoea, as described. This supports the notion that cognitive dysfunction in NAFLD is multifactorial.

## 5. Clinical Considerations and Future Perspectives

The current studies suggest that NAFLD patients incur cognitive dysfunction, which constitutes a multifactorial, metabolic encephalopathy with features of both hepatic and diabetic encephalopathy—intertwined with the adverse effects of the metabolic syndrome (Figure 1). Moreover, in the assessment of patients, it is important to acknowledge that NAFLD is also associated with a range of nonspecific physical and mental symptoms that can indirectly influence cognitive function [18,19]. Specifically, fatigue, that is a common complaint among patients with NAFLD and other liver diseases, can affect attention, concentration, and working memory [19,150,151,152]. In addition, NAFLD patients more frequently suffer from neuropsychiatric symptoms such as anxiety and depression that are also associated with impaired memory, attention, and executive function [153,154,155].

Importantly, the cognitive phenotype of NAFLD has yet to be thoroughly characterized, and as such, the psychometric and neuropsychological test batteries used in previous studies do not sufficiently address the broad metabolic encephalopathy implied by current evidence. It follows that our validated tests for hepatic encephalopathy will probably not suffice in the characterization of NAFLD patients either. To cover the full spectrum of NAFLD cognitive dysfunction, and until focused NAFLD batteries have been validated, we need to broaden our test batteries by including, as a minimum, a test of learning and/or working memory such as the comprehensive repeatable battery for the assessment of neuropsychological status (RBANS) [156] as well as screen for confounding co-morbidities.

In addition to improving diagnostic tools to correctly identify this new metabolic encephalopathy, future research should focus on the impacts of it on daily living, and on its pathological mechanisms, so as to elucidate treatment options and determine reversibility. In particular, longitudinal studies of biopsy-proven NAFLD are warranted to assess the gravity of cognitive dysfunction in this disease, including the impact of liver disease severity (NASH and fibrosis) and systemic inflammation. The majority of evidence from animal studies highlights neuroinflammation as the predominant facilitator of cognitive dysfunction in the metabolic syndrome and NAFLD, and this requires confirmation in mechanistic human studies. Furthermore, whilst direct therapeutical intervention for cognitive dysfunction seems premature at this stage, life-style changes and bariatric surgery focused on weight reduction are likely to revert the neurobehavioral changes described in this review. However, this notion needs to be confirmed in intervention studies where cognition is carefully characterized.

The cognitive and functional difficulties observed in NAFLD come with increased healthcare resources and profound societal implications including reduced quality of life and work productivity, need for care, and compliance with treatment regimens [157]. This represents a significant burden of illness and emphasizes the unmet needs of these patients, and thus, healthcare professionals should carefully consider the mental and physical quality of life of patients with NAFLD, even for those who do not have advanced liver disease. The above highlights the need for effective interventions, and whilst no pharmaceutical therapies are currently approved for the treatment of NAFLD, numerous drugs are in the pipeline. Some of these drugs, e.g., glucagon-like peptide-1 (GLP-1) analogues and peroxisome proliferator-activated receptor (PPAR) agonists exhibit well-documented neuroprotective, neurotrophic and anti-inflammatory effects, which may be taken into account when considering treatment options for NAFLD in the future [158,159].

## Figures and Tables

**Figure 1 jcm-10-00673-f001:**
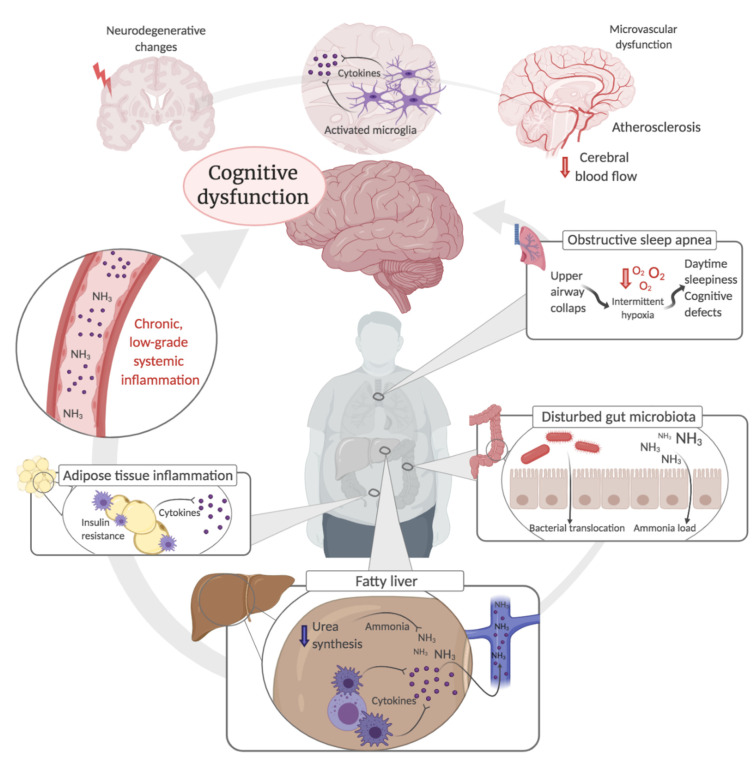
Possible mechanisms contributing to cognitive dysfunction in non-alcoholic fatty liver disease.

**Table 1 jcm-10-00673-t001:** Evidence for cognitive dysfunction in non-alcoholic fatty liver disease.

	Design and Study Population	Controlling for Important Confounders	Diagnosis of NAFLD and Fibrosis	Neuropsychological Tests	Cognitive Domains Assessed	Main Findings	Conclusion	Important Limitations
**Felipo 2012** **(Spain)** [38]	Cross-sectional. 40 NAFLD (*n* = 29 steatosis/*n* = 11 NASH), 54 controls.	None.	Liver biopsy.	Digit Symbol Substitution Test (DST).	Visuospatial function and psychomotor speed.	5/11 Patients with pre-cirrhotic NASH were classified as having minimal hepatic encephalopathy (MHE) on the PSE-test ^1^ and performed poorly on the NCT-A and NCT-B, LTT (all *p* < 0.001), and SDT (*p* < 0.01), compared with healthy controls. NASH subgroup with MHE had higher levels of ammonia and IL-6 compared to other NASH, NAFLD, and controls.	Suggests MHE-related cognitive deficits in pre-cirrhotic NASH, but not simple steatosis.	Small sample size with subgroup analysis. All NAFLD patients undergoing surgery for morbid obesity (no diabetes status). Raw data on cognitive tests missing.
				Trailmaking A test (NCT-A).	Attention and psychomotor speed.			
				Trailmaking B test (NCT-B).	Executive function.			
				Serial Dotting Test (SDT).	Attention and working memory.			
				Line Tracing Test (LTT).	Visuospatial function.			
**Seo 2016** **(USA)** [30]	Cross-sectional, population-based. 874 NAFLD, 3598 controls.	Age, education, diabetes, BMI, cardiovascular disease.	Ultrasound. NAFL fibrosis score (NFS *).	Simple Reaction Time Test (SRTT).	Psychomotor speed.	NAFLD patients had poor performance on the SDLT (β, 95% CI: 0.105 to 1.347) and also worse performance on the SRTT and SDST, but non-significantly so after adjusting for life-style related confounders (β, 95% CI: −0.496 to 14.679; −0.009 to 0.211). Poor performance on the SDST and SDLT scores were associated with increasing blood transaminases.	Suggests problems with memory and attention in NAFLD.	No biopsy-proven NAFLD. Persons aged > 59 years not included.
				Digit Symbol Substitution Test (SDST).	Visuospatial function and psychomotor speed.			
				Serial Digit Learning Test (SDLT).	Memory and attention.			
**Takahashi 2017 (Japan)** [39]	Cross-sectional. 24 female NAFLD, 15 age-matched controls.	None.	Ultrasound.	Verbal Fluency Task (VFT).	Executive function, verbal fluency.	NAFLD patients performed significantly worse on the VFT than controls, listing on average 2 words fewer during the test (*p* = 0.03).	Suggests problems with executive function and semantic fluency in NAFLD.	Small sample size, no adjustment for confounding. No biopsy-proven NAFLD. Limited cognitive assessment.
**Tuttolomondo 2018 (Italy)** [34]	Cross-sectional. 83 NAFLD (7,5% cirrhosis, 52% NASH), 80 controls.	Age, diabetes, BMI, cardiovascular disease.	Liver biopsy (in 65%). Ultrasound, liver stiffness (transient elastography).	Mini Mental State Examination (MMSE) ^2^.	Visuospatial function, executive function, memory, attention, language, and orientation.	NAFLD group performed worse on the MMSE than controls, independent of confounders (mean ± SD, 26.9 ± 1.6 vs. 28.0 ± 1.4; *p* < 0.0001). In NASH patients, poor performance on the MMSE ^2^ was associated with ballooning (β, 95% CI: −2.65 to −0.037; *p* = 0.044). No difference between NASH vs. non-NASH or low fibrosis vs. high fibrosis.	Suggests global reduction of cognitive function in NAFLD.	Small sample size. Limited cognitive assessment.
**Filipovic 2018 (Serbia)** [35]	Cross-sectional. 40 NAFLD, 30 controls with functional dyspepsia or irritable bowel syndrome.	Age, diabetes equally distributed between groups, but not otherwise controlled for.	Ultrasound (+ elevated ALT or AST).	Montreal Cognitive Assessment (MoCA) ^3^.	Visuospatial function, executive function, memory, attention, language, and orientation.	MoCA score was lower in NAFLD patients (mean ± SD, 24.07 ± 3.18 vs. 27.17 ± 2.35; *p* < 0.001), and NAFLD patients had a 4-fold increased risk of having an abnormal MoCA ^3^ score, compared with controls (RR, 95% CI: 1.815 to 8.381; *p* = 0.0005).	Suggests global reduction of cognitive function in NAFLD.	Small sample size, no adjustment for confounding. No biopsy-proven NAFLD.
**Celikbilek 2018 (Turkey)** [36]	Cross-sectional. 70 NAFLD, 73 controls.	Age, education, diabetes, metabolic syndrome.	Ultrasound, FIB-4 score **.	Montreal Cognitive Assessment (MoCA) ^3^.	Visuospatial function, executive function, memory, attention, language, and orientation.	NAFLD was associated with lower MoCA score on univariate regression analysis (OR = 2.99; *p* = 0.002), but not after adjusting for confounders (multivariate). MoCA score was negatively correlated with FIB-4 ** score.	Suggests global reduction of cognitive function in NAFLD (mostly executive and visuospatial function).	No biopsy-proven NAFLD. Patients with morbid obesity not included.
**Weinstein 2018 (USA)** [31]	Cross-sectional, population-based. 413 NAFLD (174 +T2DM), 689 controls (142 +T2DM). Age > 60 years.	Age, education, obesity, cardiovascular disease. Diabetes controlled for in subgroup analysis.	Presence of fatty liver index score *** ≥ 60.	Consortium to Establish a Registry for Alzheimer Disease – Word Learning subset (CERAD-WL).	Verbal memory (immediate and delayed recall).	NAFLD patients without T2DM did not demonstrate cognitive dysfunction, but NAFLD + T2DM performed worse than T2DM only and healthy controls on the DSST (mean ± SE, 47.1 ± 1.7 vs. 56.0 ± 1.1 and 53.6 ± 1.2). NAFLD + T2DM was associated with poor performance on DSST after adjusting for confounders (β, 95% CI: −6.75 to −0.12; *p* < 0.01).	Suggests no specific cognitive impairments in NAFLD.	No biopsy-proven NAFLD. Not generalizable to younger individuals.
				Animal Fluency Test (AFT).	Executive function, verbal fluency.			
				Digit Symbol Substitution Test (DSST).	Visuospatial function, psychomotor speed.			
**An 2019** **(USA)** [37]	Cross-sectional. 23 NAFLD, 21 sex-matched controls.	None. 8/23 NAFLD patients with diabetes.	Liver biopsy (2/23 by transient elastography).	The Repeatable Battery for the Assessment of Neuropsychological Status (RBANS) ^4^.	Immediate and delayed memory, attention, language, and visuospatial memory.	Mean RBANS total score for NAFLD patients was below mean, but within the normative range after adjusting for age and educational level.	Suggests no specific cognitive impairments in NAFLD.	No control group for cognitive assessment. Small sample size, no adjustment for confounding.
**Weinstein 2019 (USA)** [33]	Cross-sectional, population-based. 378 NAFLD, 1278 total.	Age, education, diabetes, BMI, cardiovascular disease.	Multi-detector CT and NAFLD fibrosis score (NFS *).	WAIS-R ^5^ subtest: Logical memory delayed (LMd).	Verbal memory (delayed recall).	No significant association between NAFLD and cognitive performance on any tests after adjusting for confounders, but advanced fibrosis (NFS *) was associated with poor performance on TrA – TrB (β, mean ± SE, −0.11 ± 0.05; *p* = 0.028) and SIM (β, mean ± SE, −2.22 ± 0.83; *p* = 0.009).	Suggests problems with executive function in NAFLD with fibrosis.	No biopsy-proven NAFLD.
				WAIS-R ^5^ subtest: Visual reproduction (VRd).	Visual memory (delayed recall).			
				WAIS-R ^5^ subtest: The Similarities test (SIM).	Abstract reasoning.			
				Trailmaking A – B test (TrA-TrB).	Executive function.			
				The Hooper Visual Organization Test (HVOT).	Visual perception.			

^1^ Portosystemic Systemic Encephalopathy (PSE) test: Test battery used to diagnose minimal hepatic encephalopathy (MHE), consisting of 5 tests. Measures Portosystemic Hepatic Encephalopathy Score (PHES), sum of individual test scores measured as standard deviations outside of normal range, controlled for age. PHES < −4 = MHE. ^2^ Mini Mental State Examination (MMSE): Brief cognitive screening tool for dementia and mild cognitive impairment. Score 0–30, higher score indicates better performance. MMSE score < 25 = dementia. ^3^ Montreal Cognitive Assessment (MoCA): Brief cognitive screening tool for dementia and mild cognitive impairment. Score 0−30, higher score indicates better performance. MoCA score < 26 = dementia. ^4^ Repeatable Battery for the Assessment of Neuropsychological Status (RBANS): Neurocognitive battery for detection and characterization of dementia and mild cognitive impairment. Consists of 12 subtests, yielding 5 Index scores and a total score (mean ± SD, 10 ± 3; 100 ± 15). ^5^ Wechsler Adult Intelligence Scale - Revised (WAIS-R): Intelligence quotient test designed to measure intelligence and cognitive ability in adults and older adolescents. * NFS = −1.675 + 0.037 × age (years) + 0.094 BMI (kg/m^2^) + 1.13 × impaired fasting glucose (IFG) or diabetes (yes = 1, no = 0) + 0.99 AST/ALT ratio − 0.013 × platelets (×10^9^/L] − 0.66 × albumin (g/dL) [40]. Probability for advanced fibrosis: NFS > 0.676 (low); 0.676 < NFS < −1.455 (intermediate) < −1.455; NFS < – 1.455 (high). ** FIB-4 score = (age (years) × AST (U/L))/(platelets (10^9^/L) × √ALT (U/L)). *** Fatty Liver Index Score = e^y^/(1 + e^y^) × 100, where y = 0.953 × ln(triglycerides (mg/dL)) + 0.139 × BMI (kg/m^2^) + 0.718 × ln (GGT (U/L)) + 0.053 × waist circumference (cm) – 15.745 [41].

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
