# Peer review of "Cognitive Dysfunction in Non-Alcoholic Fatty Liver Disease—Current Knowledge, Mechanisms and Perspectives"

_jcm, 2021, doi:10.3390/jcm10040673_

Round 1
Reviewer 1 Report
This is a review of cognitive impairment in NAFLD, current knowledge and proposed mechanisms. The topic is of interest however the paper should undergo major changes before it can be considered for publication.
- There is no table 1.
- I do not think that the proper type of the literature review is 'scoping review' as these aim to map boundaries of a general topic while here the research question is clear and precise. Also scoping reviews attempt to clarify key concepts, find gaps in knowledge and usually include ongoing research.
- The authors did not follow the PRISMA guidelines rigorously. The method section is too short and misses details such as how and by who the search was done, information on data extraction and variables, precise inclusion and exclusion criteria (e.g. what ages?). In addition, the flow chart includes one manuscript from 'other sources'. What is this source? Why wasn't quantitative analysis (meta-analysis) done?
- Functional impairment is out of the scope as described by the authors and as indicated by the title as well as by the search terms that do not include any term regarding that. Moreover, depression was an exclusion criteria yet is mentioned when discussing functional impairment.
- The limitation of cognitive tests are being discussed as the main barrier to better understanding the association between NAFLD and cognition (lines 419-454; 460-466). I agree that detailed cognitive battery will provide information on domains affected, but it will not be capable of teasing out the 'causes of metabolic encephalopathy'. For that, the authors should discuss other limitations of current studies including the various technologies used to assess steatosis and fibrosis and their validity and reliability.
Overall, it feels as there are two different manuscripts combined in one: a literature review and a paper that discusses potential mechanisms for the NAFLD-cognition associations. I suggest leaving out the review or improve the literature review and shorten the mechanism section.
Author Response
We thank Reviewer 1 for the critical appraisal of our manuscript. Before replying to the specific comments made, we would like to start by addressing the reviewer’s final comment:
Overall, it feels as there are two different manuscripts combined in one: a literature review and a paper that discusses potential mechanisms for the NAFLD-cognition associations. I suggest leaving out the review or improve the literature review and shorten the mechanism section.
Our Reply: This is a crucial consideration, which has also been discussed between authors while preparing the manuscript. We understand the reviewer’s comment and agree that the aim of this review was in fact two-fold: 1) To provide a detailed review of the current evidence for cognitive dysfunction in patients with NAFLD; 2) to comprehensively discuss the possible mechanisms contributing to NAFLD cognitive dysfunction. The reason for combining the two was that we found it important to discuss the pathophysiological characteristics of NAFLD in order to provide a fundamental basis for a better understanding of the clinical phenotype of NAFLD cognitive dysfunction and for discussing characterization and reversibility of this condition. The majority of the clinical studies on this topic have investigated cognitive dysfunction in NAFLD from a mechanistic perspective, thus making it natural to combine the two.
We do understand that the combination of the PRISMA-methodological scoping review, where we sought to provide the highest possible scientific standard of literature review, and the mechanistic discussion leaves the specific aim of the review somewhat unclear. Therefore, we have now changed the manuscript to a standard literature review (not scoping or systematic) and accordingly, the methods section and compliance with PRISMA guidelines has been removed. However, we have decided to keep Table 1, since it provides important information about the results and various neuropsychological tests discussed later in the review. The description of aims in the abstract and introduction has been revised (line 26-27; line 56-60). We sincerely hope that these changes have improved the overall impression of the manuscript format as requested by the reviewer.
- There is no table 1.
Our Reply: We thank the reviewer for pointing out this error, which has now been corrected. The manuscript contains only one table, Table 1.
- I do not think that the proper type of the literature review is 'scoping review' as these aim to map boundaries of a general topic while here the research question is clear and precise. Also scoping reviews attempt to clarify key concepts, find gaps in knowledge and usually include ongoing research.
Our Reply: Please see our comments above.
- The authors did not follow the PRISMA guidelines rigorously. The method section is too short and misses details such as how and by who the search was done, information on data extraction and variables, precise inclusion and exclusion criteria (e.g. what ages?). In addition, the flow chart includes one manuscript from 'other sources'. What is this source? Why wasn't quantitative analysis (meta-analysis) done?
Our Reply: Please see our comments above. However no longer relevant, we would like to answer the questions asked by the reviewer.
- The one manuscript included from another source was Takahashi 2017. This manuscript was not included from the search strategy, but was identified through reviewing the literature and was deemed fitting to the set inclusion/exclusion criteria.
- The included articles all assess cognitive function in patients with NAFLD, however, a variety of different neuropsychological tests are used. Only few studies used similar tests that would allow for quantitative analysis, and thus, a meta-analysis would not be statistically feasible.
- Functional impairment is out of the scope as described by the authors and as indicated by the title as well as by the search terms that do not include any term regarding that. Moreover, depression was an exclusion criteria yet is mentioned when discussing functional impairment.
Our Reply: We thank the reviewer for pointing out this important misunderstanding. The section on everyday living and functional impairment was intended as a supplement to the systematic review of cognitive dysfunction in NAFLD and was not part of the search strategy, as pointed out. We have changed that to a section of its own with the new heading ‘Impact of cognitive dysfunction on daily living in NAFLD’ (line 110) to further emphasize cognitive dysfunction (and not functional impairment, depression etc.) as the focus for this review. We have also shortened and rephrased the section, and replaced a few references to accommodate the research focus (line 110-128).
- The limitation of cognitive tests are being discussed as the main barrier to better understanding the association between NAFLD and cognition (lines 419-454; 460-466). I agree that detailed cognitive battery will provide information on domains affected, but it will not be capable of teasing out the 'causes of metabolic encephalopathy'. For that, the authors should discuss other limitations of current studies including the various technologies used to assess steatosis and fibrosis and their validity and reliability
Our Reply: We thank the reviewer for pointing this out. Clearly, we also do not believe that improved neuropsychological testing will help identifying causes of metabolic encephalopathy. This was poorly phrased on our behalf, and we have deleted the sentence in the revised manuscript.
Reviewer 2 Report
Overall, an excellent review on the topic area. Clear and well-written. I think the authors did a thorough job in reviewing the literature in a comprehensive manner.
Minor comments:
Table 2 gives excellent information, but it is a bit hard to read, since there is so much information included (columns become very small). Is there a way to break into two distinct tables to make it more readable?
In the supplementary materials, the Prisma flow chart needs two adjustments:
- For the articles that were reviewed at the full-text level (16), the ones that were excluded need "reasons" for exclusion. The 7 that were excluded at that stage, why were they excluded?
- The last box about meta-analysis should be removed.
Author Response
1. Table 2 gives excellent information, but it is a bit hard to read, since there is so much information included (columns become very small). Is there a way to break into two distinct tables to make it more readable?
Our Reply: We thank the reviewer for the suggestion and agree that the content in Table 1 is difficult to read. For improved readability, we have now removed the columns “Neuropsychological tests” and “Cognitive functions assessed” from the table. However, as we find this information important and of possible interest to the reader, we have added a Supplementary Table 1 outlining the applied neuropsychological tests of the studies in Table 1, including a short description of how the tests are executed (line 396-397). Should the reviewer or editor prefer to move this table to the main manuscript (as a Table 2), we welcome this.
2. In the supplementary materials, the Prisma flow chart needs two adjustments:
- For the articles that were reviewed at the full-text level (16), the ones that were excluded need "reasons" for exclusion. The 7 that were excluded at that stage, why were they excluded?
- The last box about meta-analysis should be removed.
Our Reply: We thank the Reviewer for pointing out these shortcomings in the methods section and supplementary material. Due to comments made by Reviewer 1, we have changed the form of the manuscript to a standard review, and therefore, the PRISMA checklist and flow-chart are not included in this revision.
However no longer relevant, we would like to answer the questions asked by the reviewer. We agree that the methods of the search strategy were not sufficiently described (incl. reasons for exclusion). The mentioned 7 articles were excluded if they did not fit the inclusion criteria of assessing cognitive function using objective neuropsychological and/or psychometric tests, or if the data was not compared to any form of normative data from healthy individuals or other patient groups.
Round 2
Reviewer 1 Report
The manuscript has improved however some issues remain:
- Abstract: "Recent evidence suggests NAFLD may be a cause of cognitive dysfunction independent of the contributions of the metabolic syndrome, i.e. systemic inflammation, vascular dysfunction and sleep apnoea." Both language and meaning need to be improved in this sentence. Does previous literature indeed suggest a causal association independently of those factors? also i.e. means that metabolic syndrome is inflammation, vascular dysfunction and sleep apnea.
- Abstract:Line 23: the sentence is redundant ("thus, patients with NAFLD …")
- Abstract:Line 27: regarding characterization of cognitive dysfunction see comments 5.
- line 125: the subtitle is not appropriate as the section below does not discuss how cognitive impairment in NAFLD affects daily living. Despite the authors' revision of this section, it still seems out of context and vague. It generally discusses other symptoms and conditions associated with NAFLD, some are related to cognition (e.g. sleep, depression) and others are not (e.g. frustration, fear) but it is not in one direction of NAFLD affecting these conditions as the title implies. My suggestion is that authors will consider inserting this section under possible mechanisms. This could be mentioned as indirect associations between NAFLD and cognitive function (e.g. through depressive symptoms, sleep problems) and can be added to the figure too.
- line 432: it is still not clear to me why the authors put emphasize on methods used to characterize cognition. Indeed, cognitive assessments differ between studies and it is true that a better harmonization and comprehensive assessments are needed. However, there are multiple important sources for heterogeneity between studies that the authors do not mention (e.g. control for confounders, methods for assessing NAFLD and NASH).In line 435 the authors state that no international consensus exists on the tests but this is also true for Alzheimer's disease and other dementias. Why not addition the cognitive tests from the supplemental table to table 1? That way the authors present and discuss the heterogeneity between studies in several aspects in a more balanced way.
Author Response
- Abstract: "Recent evidence suggests NAFLD may be a cause of cognitive dysfunction independent of the contributions of the metabolic syndrome, i.e. systemic inflammation, vascular dysfunction and sleep apnoea." Both language and meaning need to be improved in this sentence. Does previous literature indeed suggest a causal association independently of those factors? also i.e. means that metabolic syndrome is inflammation, vascular dysfunction and sleep apnea.
Our Reply: We agree that this sentence is unclear, and we have now rephrased it (line 16-19).
- Abstract: Line 23: the sentence is redundant ("thus, patients with NAFLD …").
Our Reply: We thank the reviewer for pointing this out, and the sentence has now been deleted from the abstract.
- Abstract: Line 27: regarding characterization of cognitive dysfunction see comments 5.
Our Reply: We have revised the discussion of characterization of cognitive dysfunction according to the reviewer’s comments.
- line 125: the subtitle is not appropriate as the section below does not discuss how cognitive impairment in NAFLD affects daily living. Despite the authors' revision of this section, it still seems out of context and vague. It generally discusses other symptoms and conditions associated with NAFLD, some are related to cognition (e.g. sleep, depression) and others are not (e.g. frustration, fear) but it is not in one direction of NAFLD affecting these conditions as the title implies. My suggestion is that authors will consider inserting this section under possible mechanisms. This could be mentioned as indirect associations between NAFLD and cognitive function (e.g. through depressive symptoms, sleep problems) and can be added to the figure too.
Our Reply: Revisiting the section, we agree that the subtitle does not correctly reflect the content, as pointed out by the reviewer, and we recognize that this section is obsolete under the scope of the current manuscript. However, and as also pointed out by the reviewer, some of these conditions do affect cognitive function as indirect associations with NAFLD. Therefore, we have moved parts of the section to ‘Clinical considerations and future perspectives’, addressing the importance of fatigue, anxiety, and depression when evaluating cognitive function in NAFLD patients (line 403-409). We did not add this topic to ‘possible mechanisms’, as otherwise suggested by the reviewer, since we consider these non-specific physical and mental symptoms as clinical complications (in line with cognitive dysfunction).
- line 432: it is still not clear to me why the authors put emphasize on methods used to characterize cognition. Indeed, cognitive assessments differ between studies and it is true that a better harmonization and comprehensive assessments are needed. However, there are multiple important sources for heterogeneity between studies that the authors do not mention (e.g. control for confounders, methods for assessing NAFLD and NASH). In line 435 the authors state that no international consensus exists on the tests but this is also true for Alzheimer's disease and other dementias. Why not addition the cognitive tests from the supplemental table to table 1? That way the authors present and discuss the heterogeneity between studies in several aspects in a more balanced way.
Our Reply: We apologize if the aim of this section is unclear. We find it essential and a natural follow-on to our review on cognitive dysfunction in NAFLD to discuss methods to characterize cognition, as the cognitive phenotype of NAFLD has yet to be thoroughly characterized. So, the section is not meant as a discussion and comparison of the different studies as such, but instead we attempt to point out and identify specific tests (for evaluation of specific cognitive domains) that can be used in future test strategies. However, we do agree that the heterogeneity in methods for assessing NAFLD and NASH is indeed an important source of varying test results in the studies discussed, as mentioned in the ‘Evidence for cognitive dysfunction in NAFLD’ section. We have now also addressed this in the present section (line 381-382). Regarding control for confounders, our discussion is only based on studies that were large enough to control for important confounders as mentioned in line 377-379 and summarized in Table 1 column “controlling for important confounders”.
Regarding the reviewer’s comments on international consensus on test strategies for NAFLD; we are aware that there is no complete international consensus on a specific test strategy for Alzheimer’s or other dementias. However, the specific cognitive deficits in these diseases, including those of hepatic encephalopathy, have been thoroughly characterized (in contrast to cognition in NAFLD as mentioned above) and highly validated test batteries exist for the specific diseases, which are clinically well-established and recommended in international guidelines. We understand, however, that the sentence wrongfully implies that international consensus on a specific test strategy exists for other diseases and therefore, we have now revised the sentence (line 367-369).
We agree that the specific neuropsychological tests and cognitive domains assessed should be included in Table 1, as in the original manuscript (please see our comments to the editorial office above).